# Early CSF Biomarkers and Late Functional Outcomes in Spinal Cord Injury. A Pilot Study

**DOI:** 10.3390/ijms21239037

**Published:** 2020-11-27

**Authors:** Rita Capirossi, Beatrice Piunti, Mercedes Fernández, Elisa Maietti, Paola Rucci, Stefano Negrini, Tiziana Giovannini, Carlotte Kiekens, Laura Calzà

**Affiliations:** 1Spinal Unit, Montecatone Rehabilitation Institute, 40026 Imola, Italy; rita.capirossi@montecatone.com (R.C.); beatrice.piunti@montecatone.com (B.P.); tiziana.giovannini@montecatone.com (T.G.); carlotte.kiekens@montecatone.com (C.K.); 2Department of Veterinary Medical Sciences, University of Bologna, 40064 Ozzano Emilia, Italy; mercedes.fernandez@unibo.it; 3Department of Biomedical and Neuromotor Sciences, University of Bologna, 40126 Bologna, Italy; elisa.maietti2@unibo.it (E.M.); paola.rucci2@unibo.it (P.R.); 4Department of Biomedical, Surgical and Dental Sciences, University of Milan, 20122 Milan, Italy; sefano.negrini@unimi.it; 5IRCCS Istituto Ortopedico Galeazzi, 20161 Milan, Italy; 6Department of Pharmacy and Biotechnology, University of Bologna, 40126 Bologna, Italy; 7Ciri-SDV, University of Bologna, 40126 Bologna, Italy

**Keywords:** spinal cord injury, CSF, biomarkers, clinical scales

## Abstract

Although, biomarkers are regarded as an important tool for monitoring injury severity and treatment efficacy, and for predicting clinical evolution in many neurological diseases and disorders including spinal cord injury, there is still a lack of reliable biomarkers for the assessment of clinical course and patient outcome. In this study, a biological dataset of 60 cytokines/chemokines, growth factorsm and intracellular and extracellular matrix proteins, analyzed in CSF within 24 h of injury, was used for correlation analysis with the clinical dataset of the same patients. A heat map was generated of positive and negative correlations between biomarkers and clinical rating scale scores at discharge, and between biomarkers and changes in clinical scores during the observation period. Using very stringent statistical criteria, we found 10 molecules which correlated with clinical scores at discharge, and five molecules, which correlated with changes in clinical scores. The proposed methodology may be useful for generating hypotheses regarding “predictive” and “treatment effectiveness” biomarkers, thereby suggesting potential candidates for disease-modifying therapies using a “bed-to-bench” approach.

## 1. Introduction

Spinal cord injury (SCI) is one of the leading causes of mortality and permanent disability in the world. According to the Global Burden of Diseases, Injuries, and Risk Factors Study (GBD) 2019 (GBD, 2019), the age-standardized incidence rate for SCI is 13 per 100,000, while the prevalence is 368 per 100,000. While, 90% of cases are due to falls or accidents. Injuries are more prevalent in males, with a peak incidence at around 30 years of age, but a growing rate of acute SCI is now seen in the elderly due to falls [1]. The direct lifetime costs of care range between $1.1 and 4.7 million per person in the US.

SCI is a broad term which encompasses various degrees of neurological deficits. Only about 1% of cases resolve completely [1], and almost 45% of SCI patients experience severe neurological deficits, in some cases, with complete or incomplete tetraplegia, with or without respiratory problems, in addiction to bowel, bladder, and sexual dysfunctions [1]. Outcomes depend on the characteristics of the lesion, pharmacological therapy, surgical interventions during the acute phase, the setting and length of hospitalization, and the type of rehabilitation services offered.

A correct assessment of SCI severity is of utmost importance for predicting functional outcomes, and for the implementation of targeted individual care plans. However, the tools available to assess the severity of spinal cord tissue destruction and to predict recovery for SCI patients remain limited. Currently, the gold standard for patient evaluation in the first 72 h postinjury is a combination of magnetic resonance imaging and American Spinal Injury Association (ASIA) Impairment Scale (AIS) [2,3]. However, such approaches have little impact on the prediction of lesion evolution and functional outcome.

Over the last few years, great research efforts have been devoted to the study of biomarkers in blood and cerebrospinal fluid (CSF) in humans and in animal models during the acute phase of traumatic SCI, as possible predictors of functional outcome. Serum neurofilament light chain concentration in plasma, for example, used in other neurological conditions, has been suggested as prognostic marker in SCI patients [4]. However, CSF is believed to better reflect the central nervous system compartment with respect to blood, due in part to the characteristics of the blood-brain barrier [5]. Its drawback include the invasiveness of the lumbar puncture needed to access the CSF. The complex nature of SCI, triggering a variety of different pathological events, such as blood coagulation, inflammation, excitotoxicity, altered redox balance, demyelination and neurodegeneration, is also difficult to capture using a single marker alone [6,7,8].

We carried out a clinical study of human CSFs, collected within 24 h of traumatic SCI, in order to measure different classes of molecules, and to test their biological properties on different types of cells (neurons and oligodendrocyte precursor cells). From the 60 cytokines, chemokines, growth factors, and structural biomarkers investigated, we identified several proteins that are transiently regulated 24 h after SCI [9]. We linked these biochemical results collected very early after lesion to the clinical outcome at patient’s discharge, in order to investigate if the biochemical profile of each patient could predict the functional outcome. In particular, we performed an exploratory correlation analysis (Spearman’s correlation coefficient) of 38 biomarkers, altered in the CSF during early spinal cord injury, with the clinical scales currently used for monitoring the neurological progression and outcome of patients with SCI in a real-world clinical setting. To the best of our knowledge, this is one of the very few studies attempting to investigate CSF biomarkers in very early SCI as predictors of functional outcome at lesion stabilization and hospital discharge.

## 2. Results

### 2.1. Participants and Biomarkers

Eleven patients were enrolled in this study. One patient was subsequently excluded because his CSF sample was grossly contaminated with blood, and one patient died about one month after injury, giving a final sample consisting of 9 patients, 8 male and 1 female. The mean age was 57.3 years (range 35–74), all with cervical or thoracic SCI with different AIS grades. The demographic and clinical characteristics of the sample are listed in Table 1. The patients were hospitalized in rehabilitation division for a median of 132 days (range 47–320), and the median number of days between trauma and admission was 3 (range 2–22).

The 38 CSF biomarkers, included in the analysis, are listed in Table 2. CSF was sampled within 24 h of injury, according to the inclusion criteria reported in the Methods.

### 2.2. Correlation Analysis

Two sets of correlation analyses were performed. The first focused on the association between biomarkers and rating scales at discharge (Table 3), the second on the association between biomarkers and changes in scale scores between baseline and discharge (Table 4).

#### 2.2.1. Neurological Level

Seven patients showed an increase in neurological functioning levels during the hospital stay and two of them showed an improvement in AIS grade. The neurological level at discharge was positively associated with MIP-1β (r = 0.763), MIP-1α (r = 0.644), MCP-1 (r = 0.559), IL-9 (r = 0.661) and IL-18 (r = 0.509). The change in neurological level since admission was positively correlated to MIP-1α (r = 0.508) and IL-9 (r = 0.561).

#### 2.2.2. Light Touch Score

The median score was 66 (range 40–106) at admission and 76 (range 45–106) at discharge. There was a positive correlation between the score at discharge and PDGF-AA (r = 0.800), BDNF (r = 0.700), t-TAU (r = 0.762), p-TAU (r = 0.650), while there was a negative correlation with IFN-α2 (r = −0.850), SCF (r = −0.600), IL-13 (r = −0.550) and NSE (r = −0.517).

Score variation was instead negatively correlated with MPO (r = −0.812), RANTES (r = −0.653) and MIG (r = −0.527).

#### 2.2.3. Pin-Prick Score

The median score was 64 (range 18–112) at admission and 75 (range 18–112) at discharge. There was a positive correlation between the score at discharge and PDGF-AA (r = 0.717), BDNF (r = 0.683), t-TAU (r = 0.644), p-TAU (r = 0.550), and a negative correlation with IFN-α2 (r = −0.650), IL-13 (r = −0.617) and SCF (r = −0.517).

Score variation was negatively correlated to IL-18 (r = −0.576) and sVCAM (r = −0.505).

#### 2.2.4. Motor Score

The median motor score was 50 (range: 0–79) at admission and 50 (range: 0–93) at discharge. The score at discharge was positively correlated with p-TAU (r = 0.763), t-TAU (r = 0.613), NCAM (r = 0.746), PDGF-AA (r = 0.712), Cathepsin-D (r = 0.695) and BDNF (r = 0.678); while it was negatively correlated with MCP-1 (r = −0.746), MIP-1α (r = −0.661), IFN-α2 (r = −0.644), IL-15 (r = −0.610), IL-8 (r = −0.593), IL-9 (r = −0.559), IL-6 (r = −0.526) and TNF-α (r = −0.526).

The score variation was positively correlated with p-TAU (r = 0.514), PDGF-AA (r = 0.653) and BDNF (r = 0.557) and negatively correlated with IL-9 (r = −0.827), MIP-1α (r = −0.870), MIP-1β (r = −0.748), MCP-1 (r = −0.740), IL-15 (r = −0.635), IL-16 (r = −0.522) and IL-8 (r = −0.505).

#### 2.2.5. SCIM

The median SCIM score was 6 (range: 0–20) at admission and 44 (range: 2–96) at discharge. SCIM at discharge was positively correlated to Cathepsin-D (r = 0.867), p-TAU (r = 0.700), NCAM (r = 0.583) and MPO (r = 0.500) while it was negatively correlated to IL-2rα (r = −0.500).

SCIM variation between admission and discharge was positively correlated with MPO (r = 0.832) RANTES (r = 0.672) and Cathepsin-D (r = 0.622), and negatively correlated with IL-2rα (r = −0.630).

#### 2.2.6. MAS Upper Limbs

At admission, no patient reported an increase in upper extremities muscle tone, MAS score was 0 for each patient. Only two patients reported an increase in muscle tone during hospitalization and at discharge. The upper limbs MAS score at discharge was positively correlated with IFN-α2 (r = 0.517) and negatively correlated with PDGF-AA (r = −0.725), t-TAU (r = −0.676), p-TAU (r = −0.518), MIG (r = −0.621), BDNF (r = −0.518) and RANTES (r = −0.518).

#### 2.2.7. MAS Lower Limbs

Two patients reported an increase in lower extremities muscle tone at admission and six at discharge. The lower limbs MAS score at discharge was positively correlated to MIF (r = 0.598) and NSE (r = 0.572) and negatively associated with NGF-β (r = −0.743), sICAM (r = −0.667), IL18 (r = −0.624), SDF-1α (r = −0.633), PDGF-AB/BB (r = −0.537), BDNF (r = −0.520) and MIG (r = −0.511). The score variation between admission and discharge was negatively associated with sVCAM (r = −0.694), IL-18 (r = −0.575) and SCGF-β (r = −0.575).

## 3. Discussion

The search for reliable biomarkers in biological fluids (blood and CSF), able to provide an objective and accurate diagnosis of neurological defects, predict functional outcome and/or monitor the efficacy of therapies, is critical in the field of chronic neurodegenerative diseases, as well as brain and spinal cord injuries and disorders. The idea of using a “biological sensor” throughout the course of the disease is also hindered by the complex nature of such conditions, which evolve over years and decades. The pathological sequelae, which follow spinal cord trauma, are determined by a cascade of events evolving over days, weeks and months, triggered by the mechanical impact and consisting of mechanisms such as blood coagulation, inflammation, excitotoxicity, demyelination and neurodegeneration, all of which have the potential to generate different “cell specific” markers in biological fluids [10,11]. As a result, CSF composition reflects this pathological sequence and changes accordingly, but also reflects the wide individual heterogeneity of the lesions.

The list of proposed biomarkers, usually based on an “hypothesis-generating” approach, varies according to the aim of testing, the underlying pathology and the stage of the lesion. As part of the Spinal Trials Understanding, Design and Implementation (STUDI) initiative, for example, a list of neurochemical biomarkers has been proposed, depending on the underlying pathology (neuronal cell body injury, astrogliosis, axonal injury, necrosis, apoptosis, neuroinflammation, demyelination etc.) [12]. Other approaches are based on preclinical data from animal models of spinal cord lesion and include novel molecules, such as miRNA [13]. Some of these studies also seek to correlate biomarker level with clinical state. For example, the combination of inflammatory (IL-6, IL-8, and MCP-1) and structural neural (tau) and glial (S100β, and GFAP) biomarkers in acute SCI patients (24-h post injury) has been proposed to reflect injury, severity according to the AIS scale [7,14]. Markers that may help to distinguish between moderate, and severe SCI events, have also been proposed [14].

However, to the best of our knowledge, very few studies have attempted to investigate CSF biomarkers in very early SCI as predictors of functional outcome at lesion stabilization. In this pilot study, we explored this possibility using a “data-driven” approach, applied in a clinical setting. Although our results are based on a small number of patients (*N* = 9), the group shows good homogeneity, due to very stringent inclusion criteria in terms of post-trauma time (<24 h), trauma characteristics (cervical and thoracic SCI), surgical treatment, and pharmacological treatments prior to CSF sampling (no steroids).

The first set of correlations is between biomarkers and rating scales at discharge, and are therefore, intended as “prognostic” biomarkers. Using the most stringent statistical criteria (< = −0.7, negative correlation; > = 0.7; positive correlation), two growth factors, three intracellular proteins, and one extracellular matrix proteins were found to positively correlate with clinical scores, while three inflammatory cytokines and one growth factor correlated negatively. In particular, PDGF-AA positively correlated with the motor, tactile and pain sensory scores, and negatively correlated with the upper limb motor score. Platelet-Derived Growth Factors (PDGFs) are a family of four cystine-knot-type growth factors (PDGF-A, -B, -C and -D), which control the growth of different tissues and cell types through dimeric tyrosine kinase receptors [15]. PDGFs are one of the major mitogens and chemo-attractants for mesenchymal and glial cells. These growth factors are neuroprotective through neurovascular unit regulation, and are abundant in neurons and glial cells, together with their receptors (PDGFRs) [16], as well as promoting the proliferation and recruitment of oligodendrocyte precursor cells, the cell responsible for myelin repair [17]. BDNF positively correlates with the sensory tactile score (and with the other motor and sensory score at significance > = 0.5), while NGF-β negatively correlates with lower limb MAS. Both of these molecules belong to the neurotrophic factor family, and a neuroprotective role towards selective neural populations is attributed to both molecules. NGF is also considered a pleiotropic molecule, which affects the development, physiology and pathology of different cell types, tissues and organs [18], and promotes angiogenesis during neural lesions [19]. While, NGF content in the spinal cord dramatically declines in models of inflammatory-demyelinating disease [20].

Among the intracellular proteins, the structural protein TAU and the lysosomal proteinase cathepsin D correlate with the clinical scores (including SCIM). TAU proteins are microtubule-associated proteins, predominantly located in CNS axons, maintaining the stability of microtubules and participating in anterograde axonal transport [21]. TAU hyperphosphorylation (pTAU) is associated with axonal pathology, while pTAU is a marker for several tau-pathologies, as well as for traumatic brain injury [22]. In experimental lesions in rats, tTAU and pTAU significantly increased at 12- and 24-h post-SCI [23], an increase negatively associated with motor performance as evaluated using the Basso, Beattie, and Bresnahan locomotor rating scale at 28 days [24]. TAU abnormalities triggered by SCI are also responsible for increases in axonal damage in the spinal cord and CSF and diffuse tau pathology in the CNS [25]. In humans, CSF tau levels showed significant increases linked to injury severity, 24 h after injury [26], and our results confirm that early TAU levels may predict motor score recovery at 6 months post-injury [5]. While, the function of lysosomal proteinase cathepsin D is not entirely clear, its involvement in the autophagy, processing and/or degradation of several neuronal proteins suggests both neuroprotective and neurotoxic roles in neurodegenerative diseases and CNS lesions [27].

With regard to inflammation-related molecules, and considering strong correlations only, 13 cytokines/chemokines related to inflammation negatively correlated with the clinical scores, the strongest correlation being observed for IFN-α2 with the light-touch score, and MCP1 with the motor score. IFN-α may play a detrimental role in brain trauma, enhancing the pro-inflammatory response, while keeping astrocyte proliferation in check [28]. Monocyte chemoattractant protein-1 (MCP-1) is a strong candidate for mediating monocyte chemotaxis to the injured nervous system, and monocyte recruitment and myelin removal are delayed following spinal cord injury in mice genetically deleted for the main MCP-1 receptor [29]. On the contrary, the chemokine (MIP-1β, or CCL4) positively correlated with neurological level. MIP-1β is a potent chemoattractant for inflammatory and other types of cells, and promotes healing and homeostasis in several tissues [30].

Last, correlations of inflammation-related molecules with SCIM (> = 0.5), motor score (> = 0.7) and neural cell adhesion molecules (NCAM) were observed. The neural cell adhesion molecule is an immunoglobulin-like neuronal surface glycoprotein, which binds to a variety of other cell adhesion proteins to mediate adhesion, guidance, and differentiation during neuronal growth. Notably, the constitutive ablation of NCAM worsens locomotor function in spinal cord injury mice [31].

The second set of correlations was between biomarkers and rating scale changes between baseline and discharge, thus intended as biomarkers for “disease evolution” and possibly useful for “treatment effectiveness”. The motor score variation over the course of the observational period negatively correlated with soluble proteins related to inflammation. e.g., IL9, IP10, MIP-1α and MIP-1β. IL9 is a pleiotropic cytokine, which affects the activity of multiple cell types in the immune compartment and in the CNS [32]. Monocyte chemoattractant protein-1 (MCP-1) is a strong candidate for mediating chemotaxis of monocytes to the injured nervous system, and monocyte recruitment and myelin removal are delayed following spinal cord injury in mice genetically deleted for the main MCP-1 receptor [33]. MIP-1 are a potent chemoattractants not only for inflammatory cells, but they also promotes healing and homeostasis in several tissues [34].

In order to speculate on a possible correlation of biochemical data with injury severity, we have considered the option of comparing CRF biomarkers among patients with different pre-operative ASIA grades. Unfortunately, a small sample size does not facilitate formal statistical testing of CSF biomarker differences to be conducte across ASIA grades due to the large inter-individual variability. To address this issue, we have graphically represented the values of CSF biomarkers by ASIA grade using dot plots. In this way possible differences between ASIA grades can be visually detected. The 4 panels, in which analytes are grouped as: (a) cytokines and chemokines, (b) growth and other factors, (c) soluble cell adhesion molecules and Neurological biomarkers, (d) other Biomarkers, are provided as Appendix A. Although we submit that the proposed CSF biomarkers might be useful predictors of late functional outcomes in patients with the same level of severity, more evidence from larger samples and additional studies is needed to draw specific conclusions.

As a final note, it is worth remarking that the 9 study patients are 3 ASIA grade A, 3 grade B and 3 grade D, which suggests that the correlations reported in the paper are not biased.

The biomarkers profile, observed in the CSF, collected within 24 h from the injury, also reflects inflammation as the prevalent pathogenic mechanism at this stage of this progressive pathology, that will chronically evolve in neurodegeneration and demyelination. Inflammation is a major player in the so-called “secondary degeneration”, that expands the initial damage, sustaining the chronic evolution of the lesion [35], and a recognized, effective target for early pharmacological intervention [36,37]. However, being the traumatic lesion itself, such as the tissue and systemic reaction, a highly personalized process, appropriate biomarker discovery strategy should be based on serial sampling of biological fluids, from injury to stabilization. Since CSF serial sampling is ethically hardly acceptable, a further effort should be dedicated to the individual comparative analysis of blood and CSF biomarkers, also considering results in animal models [38,39].

## 4. Materials and Methods

### 4.1. Study Participants and Ethical Issues

Participants were enrolled consecutively between November 2012 and September 2016 at the regional hospital trauma center (Maggiore Hospital, Bologna, Italy) and then transferred to the tertiary referral institute (Spinal Unit of Montecatone Rehabilitation Institute, Imola, Italy). Inclusion criteria were: The presence of cervical and/or thoracic SCI; surgical spine stabilization with posterior approach (this allowed us to obtain the CSF sample by lumbar puncture at the beginning of the surgical procedure) and CSF collection within 12–24 h of injury; age between 18 to 74 years, and ability to provide written informed consent to participation. When patients were unable to write, a witness not related to the patient and not involved in the study signed the written consent. Patients treated with steroids, in the early acute phase, following SCI were excluded.

The study was approved by the Ethics Committees of the Bologna Local Health Trust (Azienda Unità Sanitaria Locale di Bologna, no. CE 12105, 18/01/2013) and the Sant’Orsola-Malpighi University Health Trust (Azienda Ospedaliero Universitaria S. Orsola Malpighi, notification acceptance no. 976/2013, 12/03/2013). It was also registered on ClinicalTrials.gov (NCT01861808).

### 4.2. CSF Sampling and Biochemical Analysis

The CSF was collected at Maggiore Hospital within 24 h of injury (T0) and the samples centrifuged (2000× *g*, 10 min, 4 °C). The supernatant was then aliquoted, immediately deep frozen and stored at −80 °C at the hospital laboratory. The multiparametric protein/biomarker quantification was performed using the Luminex xMAP technology and MAGPIX platform, a technique which allows the simultaneous assay of up to 50 analytes in a maximum of 50 µl of CSF. The kits, included cytokine, chemokine, growth factor, neurological disease and neurological disorder biomarkers. Three magnetic bead panels were used for biomarker testing: 1. cytokines, chemokines and growth factors; 2. neurological disorders, and 3. neurodegenerative diseases. To quantify the cytokines, chemokines and growth factors, two panels were used of 20, and 27 analytes, respectively (custom Bio-PlexPro Human Cytokine 21-plex Assay #MF0005KMII and Bio-PlexPro Human Cytokine 27-plex Assay #MF500KCAFOY, from BioRad, Hercules, CA, USA). To quantify the potential biomarkers of neurological disorders, as well as Neurone Specific Enolase (NSE), Total Neurofibrillary Tangle Protein (TAU) and Thr231 Phosphorylated Neurofibrillary Tangle Protein (pTAU), a 3-plex analyte kit (Human Neurological Disorders, HND1MAG-39K, EMD Millipore Corporation Billerica, MA, USA) was used. Finally, to quantify the biomarkers of neurodegenerative diseases, a 10-plex analyte kit (Human Neurodegenerative Disease, HNDG3MAG-36K, Millipore) was used. Samples were processed in duplicate, and the data obtained from the beads was analyzed using xPONENT 4.1 Software. Quantifications were performed based on specific standard analyte curves.

### 4.3. Outcomes/Neurological Evaluations

Patients underwent surgical spine stabilization with a posterior approach, after a neuroradiology investigation (CT and/or NMR). Following the early acute phase at the Maggiore Hospital Trauma Center, patients were transferred to the Spinal Unit of the Montecatone Rehabilitation Institute for rehabilitation. Patients were evaluated by trained physiatrists at three different time points (on admission, three months after injury, and on discharge, if occurring after 3 months of admission), using the American Spinal Injury Association (ASIA) Impairment Scale (AIS), the Spinal Cord Independent Measure (SCIM), and the Ashworth Modified Scale (MAS).

The American Spinal Injury Association (ASIA) Impairment Scale (AIS) is the international gold standard for the evaluation of spinal cord injury. The AIS is a standardized examination consisting of a myotomal-based motor examination, dermatomal-based sensory examination, and sacral area examination [2,3], and injury severity is scored on a 5 point ordinal scale. The sensory score and sensory level are determined after examining light touch and pin prick sensations within each dermatome of the body and scoring the results on a three-point scale (0 = absent, 1 = altered, 2 = normal). The sensory exam is performed on 28 points on both sides of the body, termed “key sensory points”. The maximum score for light touch and for pin prick sensations is 112, corresponding to the degree of sensitivity preserved. The motor score and motor level, in turn, refer to the results obtained from examining and recording the strength of certain key muscle functions of the extremities using a six-point scale (from 0 = absent, to 5 = normal). The maximum motor score is 100, corresponding to the degree of strength preserved. The lowest normal sensory level may be substituted in regions without readily testable myotomes (such as in the thoracic spine). The neurological level of injury refers to the most caudal segment of the spinal cord with intact sensation and antigravity muscle strength provided there is normal sensory and motor function rostrally. The AIS further classifies injuries as complete/incomplete spinal cord injuries. A complete spinal cord injury is defined as the absence of any motor or sensory function in the sacral segments S4–S5. These injuries are rated as Grade A on the AIS. Incomplete injuries are defined as those with some degree of preserved motor or sensory function in sacral area. These are rated a grade B through E on the AIS. Patients with AIS Grade B injuries have some sensory function but no motor function. AIS Grade C and D are motor incomplete injuries with respectively less than half of the muscles (Grade C) or at least half of the muscles (Grade D) having a motor grade of 3 or more below the neurologic level of injury. Patients with Grade E injuries have normal motor and sensory examinations, but still may have abnormal reflexes or other neurologic phenomena.

The Spinal Cord Independence Measure (SCIM) assesses the ability of a person with spinal cord injury (SCI) to carry out activities of daily living (ADL) [40]. It assesses independence in 19 key areas, including: self-care (six items), respiration and sphincter management (four items), and mobility (nine items). The items are scored and weighted differently and are added to obtain a total score ranging from 0 to 100, where higher scores indicate independence in most ADL.

The Modified Ashworth Scale (MAS) is used to assess muscle spasticity in clinical practice and research [41], and is rated on a 0 to 4 Likert scale, based on the amount of resistance measured by an evaluator when attempting to move a joint through an available range of motion. The scale is scored as follows [42]: 0: No increase in muscle tone; (1): Slight increase in muscle tone, with a catch and release or minimal resistance at the end of the range of motion when an affected part(s) is moved in flexion or extension; 1+: Slight increase in muscle tone, manifested as a catch, followed by minimal resistance through the remainder (less than half) of the range of motion; (2): Marked increase in muscle tone throughout most of the range of motion, but affected part(s) are still easily moved; (3): Considerable increase in muscle tone, passive movement difficult; (4): Affected part(s) rigid in flexion or extension.

### 4.4. Statistical Methods

The patients’ demographic and clinical characteristics were reported as mean±SD or absolute and percentage frequencies. Median and range values (min-max) were used to summarize the distribution of functional scales at admission and discharge, the time between trauma and hospitalization expressed in days, and length of hospital stay. The outcomes involved the scale scores at discharge and the change in score between admission and discharge. The relationship between biomarkers and outcomes was analyzed using Spearman’s correlation coefficient, which is appropriate for small samples and which ranges between −1 and +1. Positive values indicate a positive association (scale scores increase with the increase of biomarker levels), while negative values indicate a negative association (scale scores decrease with the increase of biomarker levels). Correlation coefficients with an absolute value ≥ 0.50 and ≥0.70 were considered as indicative of a strong, and a very strong association, respectively.

The correlation matrices were displayed as a superimposed heat map to facilitate interpretation. All outcome measures were scored with the same direction, so that low values correspond to poor functioning and high values to good functioning. Analysis included only those biomarkers with detectable values. Missing or unreliable values were not imputed.

Statistical analysis was conducted using IBM SPSS version 25.0 and Stata statistical software (StataCorp. 2017. Stata Statistical Software: Release 15. College Station, TX: StataCorp LLC).

## 5. Conclusions

While, our small sample size does not permit any definitive conclusions about biomarker candidates in spinal cord injury, we consider our methodology to be a positive step in this direction. We need to investigate the relationship between biomarkers and rating scales routinely used in clinical practice, in an attempt to identify biomarkers able to predict patient outcomes and/or treatment effectiveness. As a very preliminary suggestion, we could speculate that inflammation-related markers might correlate with the clinical outcome, but the high individual variability of inflammatory reaction to trauma imposes large scale studies to drow definitive conclusions. This approach also offers a “bed-to-bench” outlook in the translational research for spinal cord injury, as well as suggesting potential candidates for disease-modifying therapies from clinical data who might otherwise be overlooked. PDGF-AA, for example, correlated with four out of seven investigated clinical scales, making it a more robust candidate than the well-established pTAU (2 out of 7).

## Figures and Tables

**Table 1 ijms-21-09037-t001:** Characteristics of study participants: sex, age, comorbidities, trauma associated with the event cause of SCI, neurological level of injury (NIL) and AIS score at T0.

ID	Gender	Age	A.I.S. Grade	Neurological Level	Comorbidities	Associated Injuries
3	Male	68	A	T6	metabolic, cardiological	thoracic
4	Male	35	B	T4	none	thoracic
5	Male	74	D	C5	metabolic, gastrointestinal	none
6	Male	42	A	T7	hematological	thoracic
7	Male	73	D	C5	metabolic, cardiological, gastrointestinal	scalp
9 *	Female	41	A	T8	none	brain, hip
10	Male	63	D	C5	osteoarticular, metabolic	brain, abdominal, thoracic
13 **	Male	53	A	C4	cardiological, metabolic, respiratory	thoracic
14	Female	53	B	T11	none	brain
16	Male	53	B	C4	psychological	brain
17	Male	55	A	T3	none	brain, thoracic

AIS = ASIA Impairment Score. A = motor and sensory complete SCI; B = motor complete and sensory incomplete SCI; C = Motor incomplete SCI (key muscle functions below the single neurological level of injury (NLI) have a muscle grade less than 3); D = Motor incomplete SCI (at least half of key muscle functions below the NLI have a muscle grade of 3 or greater). C = Cervical; T = Thoracic. * = later excluded because of glossy blood contamination. ** = later excluded because of death.

**Table 2 ijms-21-09037-t002:** List of analytes included in the correlation analysis.

Cytokines and Chemokines	CTACK, IFN-α2, IL-13, IL-15, IL-16, IL-18, IL-2r-α, IL-6, IL-7, IL-8, IL-9, IP-10, MCP-1, MIP-1α, MIP-1β, RANTES, TNF-α, TRAIL
Growth and other factors	BDNF, GRO-α, HGF, NGF-β, PDGF-AA, PDGF-AA/BB, SCF, SCGF-β, SDF-1α
Soluble cell adhesion molecules	sVCAM, sICAM, NCAM
Neurological biomarkers	NSE, p-TAU, t-TAU
Other biomarkers	MIG, MIF, cathepsin D, MPO, PAI-1

**Table 3 ijms-21-09037-t003:** Spearman’s correlation coefficients between biomarkers and rating scales at discharge. The heat map is in red for positive correlations and green for negative correlations. The color saturation is proportional to the strength of the associations between variables.

Biomarker	Neurological Level	Light Touch Score	Pin Prick Score	Motor Score	SCIM	MAS Upper Limbs	MAS Lower Limbs
CTACK	−0.390	−0.267	−0.167	0.170	0.117	0.000	−0.234
IFN-α2	0.068	−0.850	−0.650	−0.644	−0.483	0.518	0.061
IL-13	−0.339	−0.550	−0.617	−0.153	−0.067	0.207	0.017
IL-15	0.356	−0.333	−0.267	−0.610	−0.250	0.207	0.234
IL-16	0.424	−0.167	0.000	−0.356	0.000	0.104	0.043
IL-18	0.509	−0.083	0.167	−0.085	−0.017	0.104	−0.624
IL-2rα	−0.271	−0.233	−0.267	−0.085	−0.500	0.104	−0.407
IL-6	0.170	−0.300	−0.133	−0.526	−0.367	0.104	−0.052
IL-7	0.197	0.067	0.135	−0.180	−0.420	−0.365	−0.319
IL-8	0.203	−0.217	-0.167	−0.593	−0.367	0.000	0.260
IL-9	0.661	−0.200	−0.167	−0.559	−0.083	0.207	0.321
IP-10	−0.187	−0.033	−0.067	−0.407	-0.383	−0.414	−0.009
MCP-1	0.559	−0.300	−0.267	−0.746	−0.450	0.414	0.234
MIP-1α	0.644	−0.267	−0.250	−0.661	−0.250	0.207	0.303
MIP-1β	0.763	−0.233	−0.100	−0.356	0.133	0.311	0.217
TNF-α	0.237	−0.267	−0.150	−0.526	−0.267	0.104	0.113
TRAIL	0.392	−0.268	0.008	−0.247	−0.142	0.052	−0.344
RANTES	−0.220	0.217	0.450	0.305	0.250	−0.518	−0.277
BDNF	−0.220	0.700	0.683	0.678	0.417	−0.518	−0.520
NGF-β	0.060	0.126	0.193	−0.145	−0.361	−0.365	−0.743
GRO-α	0.390	0.100	0.100	−0.288	0.100	0.000	0.095
HGF	−0.017	−0.367	−0.183	−0.136	0.033	−0.104	0.052
PDGF-AA	−0.373	0.800	0.717	0.712	0.333	−0.725	−0.425
PDGF-AB/BB	−0.187	0.200	0.283	0.051	−0.033	−0.414	−0.537
SCF	−0.034	−0.600	−0.517	−0.475	−0.450	0.207	0.104
SCGF-β	−0.085	−0.417	−0.350	−0.017	0.250	0.207	−0.156
SDF-1α	0.034	0.150	0.217	0.000	−0.183	−0.311	−0.633
sVCAM	−0.119	−0.133	−0.017	−0.119	−0.267	−0.414	−0.494
NCAM	−0.034	0.433	0.433	0.746	0.583	−0.311	−0.295
sICAM	0.017	−0.233	−0.150	−0.170	−0.167	0.207	−0.667
NSE	0.119	−0.517	−0.450	−0.475	−0.083	0.311	0.572
t-TAU	−0.051	0.762	0.644	0.613	0.469	−0.676	−0.109
p-TAU	−0.187	0.650	0.550	0.763	0.700	−0.518	0.130
PAI-1	−0.017	−0.233	−0.167	−0.424	−0.233	0.000	−0.113
Cathepsin-D	−0.034	0.383	0.333	0.695	0.867	−0.104	0.173
MIF	0.254	−0.117	−0.067	−0.153	0.333	0.104	0.598
MIG	−0.441	0.200	0.417	0.305	0.017	−0.621	−0.511
MPO	0.119	0.200	0.467	0.339	0.500	−0.311	0.009

Legend:   ≤−0.7  
  ≤−0.5    ≤−0.3    ≥0.3    ≥0.5    ≥0.7  .

**Table 4 ijms-21-09037-t004:** Spearman’s correlation coefficients between biomarkers and changes in scale scores between baseline and discharge. The heat map is in red for positive correlations and green for negative correlations. The color saturation is proportional to the strength of associations between variables.

Biomarker	Neurological Level	Light Touch Score	Pin Prick Score	Motor Score	SCIM	MAS Lower Limbs
CTACK	−0.330	−0.209	0.231	0.340	0.185	−0.164
IFN-α2	−0.267	0.017	−0.169	−0.400	−0.227	−0.265
IL-13	0.134	0.226	−0.053	0.009	−0.261	−0.475
IL-15	0.312	0.067	−0.106	−0.635	−0.017	−0.201
IL-16	0.312	−0.276	−0.222	−0.522	0.311	−0.274
IL-18	0.267	−0.352	−0.576	−0.392	0.177	−0.575
IL-2rα	−0.339	0.368	−0.195	0.174	−0.630	−0.110
IL-6	−0.045	−0.126	−0.177	−0.400	0.067	−0.155
IL-7	0.143	0.126	0.053	−0.505	−0.067	−0.009
IL-8	0.238	0.190	−0.380	−0.290	−0.386	−0.295
IL-9	0.561	0.176	0.124	−0.827	−0.017	−0.146
IP-10	0.223	0.201	0.036	−0.218	−0.118	−0.201
MCP-1	0.080	0.343	0.248	−0.740	−0.269	0.164
MIP-1α	0.508	0.343	0.204	−0.870	−0.210	−0.119
MIP-1β	0.454	−0.042	0.169	−0.749	0.160	−0.164
TNF-α	0.134	−0.075	−0.142	−0.479	0.084	−0.155
TRAIL	0.018	−0.231	−0.058	−0.402	0.122	−0.321
RANTES	−0.232	−0.653	0.089	0.313	0.672	−0.055
BDNF	−0.036	−0.352	0.018	0.557	0.437	0.046
NGF-β	0.126	0.156	−0.009	−0.158	−0.195	−0.359
GRO-α	0.445	−0.025	0.186	−0.470	0.277	−0.082
HGF	0.169	−0.301	−0.257	−0.183	0.210	−0.456
PDGF-AA	−0.027	−0.226	0.018	0.653	0.303	0.119
PDGF-AB/BB	−0.054	−0.176	0.222	0.131	0.277	−0.173
SCF	0.036	0.126	−0.488	−0.313	−0.370	−0.383
SCGF-β	0.241	−0.176	−0.053	−0.070	0.219	−0.575
SDF-1α	0.018	0.084	0.186	−0.035	−0.076	−0.210
sVCAM	0.294	−0.059	−0.505	−0.131	−0.118	−0.694
NCAM	0.267	−0.352	−0.364	0.392	0.294	−0.256
sICAM	−0.062	0.025	0.062	−0.096	−0.034	−0.383
NSE	0.089	−0.034	0.018	−0.400	0.067	−0.055
t-TAU	0.336	−0.084	0.196	0.319	0.270	0.101
p-TAU	0.160	−0.276	0.257	0.514	0.471	0.219
PAI-1	0.196	−0.092	−0.275	−0.296	0.109	−0.374
Cathepsin-D	0.205	−0.418	0.177	0.383	0.622	0.073
MIF	0.276	−0.243	0.204	−0.313	0.445	0.064
MIG	−0.401	−0.527	0.009	0.470	0.445	−0.100
MPO	−0.009	−0.812	−0.098	0.104	0.832	−0.055

Legend:  ≤−0.7 
 ≤−0.5  ≤−0.3  ≥0.3  ≥0.5  ≥0.7 .

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
