# Peer review of "Early CSF Biomarkers and Late Functional Outcomes in Spinal Cord Injury. A Pilot Study"

_ijms, 2020, doi:10.3390/ijms21239037_

Round 1

Reviewer 1 Report

Interesting manuscript evaluating the potential role of early CSF biomarkers in predicting the late functional outcomes in spinal cord injury.

The sample is quite small, as recognized by the authors.

The authors could more clearly underline the novelty of their findings when compared with other papers on this topic.

Was there any difference in the CSF biomarkers between patients with different pre-operative ASIA grade?

If we consider patients with the same preoperative ASIA grade, e.g. only preoperative ASIA grade A patients, could the proposed CSF biomarkers be useful to predict the late functional outcomes?

The same evaluations could be performed for the other rating scales.

Author Response

Review 1

Interesting manuscript evaluating the potential role of early CSF biomarkers in predicting the late functional outcomes in spinal cord injury.

We thank the reviewer for this encouraging comment, and for all other suggestions

The sample is quite small, as recognized by the authors.

The authors could more clearly underline the novelty of their findings when compared with other papers on this topic.

We have better highlighted this point in the introduction, also in view of our previously published results, where the following statement (lines 63-70 original submission) ….

“We therefore carried out a clinical study of human CSF collected following traumatic SCI, given its suitability for the investigation of inflammation, neurodegeneration and re-myelination, and we also tested its biological properties on different types of cells [9]. Of the 60 cytokines, chemokines, growth factors, and structural biomarkers investigated, we identified 19 proteins which are transiently regulated 24 h after SCI. In this study, we linked biological data to clinical data to perform an exploratory correlation analysis of 38 biomarkers altered in the CSF during early spinal cord injury with the clinical scales currently used for monitoring the neurological progression and outcome of SCI patients in a real-world clinical setting”.

Will be changed to …..

“We therefore carried out a clinical study of human CSFs collected within 24hours of traumatic SCI, in order to measure different classes of molecules, and to test their biological properties on different types of cells (neurons and oligodendrocyte precursor cells). Of the 60 cytokines, chemokines, growth factors, and structural biomarkers investigated, we identified several proteins which are transiently regulated 24 h after SCI [9]. In the attempt to investigate if the biochemical profile of each patient could predict the functional outcome, we linked these biochemical results collected very early after lesion to the clinical outcome at patient’s discharge. In particular, we performed an exploratory correlation analysis (Spearman’s correlation coefficient) of 38 biomarkers altered in the CSF during early spinal cord injury with the clinical scales currently used for monitoring the neurological progression and outcome of patients with SCI in a real-world clinical setting. To the best of our knowledge, this is one of the very few studies attempting to investigate CSF biomarkers in very early SCI as predictors of functional outcome at lesion stabilization and hospital discharge.”

Was there any difference in the CSF biomarkers between patients with different pre-operative ASIA grade?

In response to the reviewer’s comment, we have considered the option to compare CRF biomarkers among patients with different pre-operative ASIA grades. Unfortunately, the small sample size does not allow to conduct formal statistical testing of CSF biomarker differences across ASIA grades due to the large inter-individual variability.

To address in a descriptive way this issue, we have graphically represented the values of CSF biomarkers by ASIA grade using dot plots. In this way possible differences between ASIA grades can be visually detected. The 4 figures, in which analytes are grouped as a) Cytokines and chemokines, b) Growth and other factors, c) Soluble cell adhesion molecules and Neurological biomarkers, d) Other Biomarkers, are provided as supplementary materials.

If we consider patients with the same preoperative ASIA grade, e.g. only preoperative ASIA grade A patients, could the proposed CSF biomarkers be useful to predict the late functional outcomes?

Although we submit that the proposed CSF biomarkers might be useful predictors of late functional outcomes in patients with the same level of severity, more evidence from larger samples and additional studies is needed to draw specific conclusions.

As a final note, it is worth remarking that the 9 study patients are 3 ASIA grade A, 3 grade B and 3 grade D, which suggests that the correlations reported in the paper are not biased.

The same evaluations could be performed for the other rating scales.

Reviewer 2 Report

IJMS-1000309

Review.

GENERAL COMMENTS:

  • This is a broad approach study of 60 markers arbitrarily selected as potentially representative of neurotrauma in SCI patients which is commendable and this is an important study.
  • The sampling of the CSF is commendable as the subarachnoid space serves as the sink to dump and dilute soluble molecules from the spinal cord. It would be useful to run the same analyses in the blood plasma with an eye of possible replacing of CSF collection for the analysis by blood collection.
  • The restriction of the sample collection and analysis to the first 24 hrs post-SCI is of doubtful value because:
  1. Recently elucidated (https://doi.org/10.1371/journal.pone.0226584. in the rat model) pathogenesis of SCI indicates that a traumatic event initiates a severe destructive inflammation that has a very long course.
  2. While the traumatic event occurs by accident therefore it is unavoidable, acute trauma to the spinal cord with resulting hemorrhage, ischemia, necrosis and edema are already there before a patient is admitted, diagnosed and a treatment initiated. There is no treatment for necrosis in the SCI other than neuroregeneration that does not yet exist.
  3. It turns out the real target in the SCI, a very chronic disease, is destructive inflammation. Effective anti-inflammatory agents administered until the inflammation is not only inhibited but also eliminated should theoretically act as neuroprotectants.
  4. In experimental rats the inflammation (macrophage infiltration in the cavity of injury, COI) can be observed and counted in histologic analysis. doi:10.3390/biomedicines8100372.; org/10.5114/fn.2019.83830  but this cannot be done in patients where in vivo analyses are required.  Here comes the significance of developing analyses of CSF/plasma to monitor the effectiveness of an anti-inflammatory treatment in clinical setting.
  • I would recommend that this paper indicates its initial role in addressing the above points and accordingly shows the direction for development of diagnostic analyses of body fluids from SCI patients in a systematic, repetitious fashion designed to monitor the success or failure of an anti-inflammatory treatment. For now it is a theoretical consideration as the elucidation of the pathogenesis of the SCI was published only recently and anti-inflammatory drugs have not entered the clinical trials yet.  However, knowing what are the dynamic curves of levels of individual cytokines, chemokines, markers of the CNS damage, etc., taken systematically from untreated patients for 4-5 months will be of very great importance to analyze the effectiveness of candidate anti-inflammatory drugs.

Author Response

Review 2

GENERAL COMMENTS:

  • This is a broad approach study of 60 markers arbitrarily selected as potentially representative of neurotrauma in SCI patients which is commendable and this is an important study.

We thank the reviewer for this encouraging comment, and for all other suggestions

  • The sampling of the CSF is commendable as the subarachnoid space serves as the sink to dump and dilute soluble molecules from the spinal cord. It would be useful to run the same analyses in the blood plasma with an eye of possible replacing of CSF collection for the analysis by blood collection.

We completely agree with this comment. We currently use the comparison of blood and CSF biomarkers in animal studies also in different animal models. Unfortunately, in this study the blood was not included in the ethic committee authorization for logistic constraints related to the patient flow in the trauma hub-and-spoke system in Emilia-Romagna region.

  • The restriction of the sample collection and analysis to the first 24 hrs post-SCI is of doubtful value because:
  1. Recently elucidated (https://doi.org/10.1371/journal.pone.0226584. in the rat model) pathogenesis of SCI indicates that a traumatic event initiates a severe destructive inflammation that has a very long course.
  2. While the traumatic event occurs by accident therefore it is unavoidable, acute trauma to the spinal cord with resulting hemorrhage, ischemia, necrosis and edema are already there before a patient is admitted, diagnosed and a treatment initiated. There is no treatment for necrosis in the SCI other than neuroregeneration that does not yet exist.
  3. It turns out the real target in the SCI, a very chronic disease, is destructive inflammation. Effective anti-inflammatory agents administered until the inflammation is not only inhibited but also eliminated should theoretically act as neuroprotectants.
  4. In experimental rats the inflammation (macrophage infiltration in the cavity of injury, COI) can be observed and counted in histologic analysis. doi:10.3390/biomedicines8100372.; org/10.5114/fn.2019.83830  but this cannot be done in patients where in vivo analyses are required.  Here comes the significance of developing analyses of CSF/plasma to monitor the effectiveness of an anti-inflammatory treatment in clinical setting.
  • I would recommend that this paper indicates its initial role in addressing the above points and accordingly shows the direction for development of diagnostic analyses of body fluids from SCI patients in a systematic, repetitious fashion designed to monitor the success or failure of an anti-inflammatory treatment. For now it is a theoretical consideration as the elucidation of the pathogenesis of the SCI was published only recently and anti-inflammatory drugs have not entered the clinical trials yet.  However, knowing what are the dynamic curves of levels of individual cytokines, chemokines, markers of the CNS damage, etc., taken systematically from untreated patients for 4-5 months will be of very great importance to analyze the effectiveness of candidate anti-inflammatory drugs.

We completely agree also with these comments and general indications. In particular, we agree on the fact that and effective pharmacological control of destructive inflammation in very early SCI results in an indirect neuroprotection. According to the reviewer’s indication, we have included the following statement at the end of the discussion:

“The biomarkers profile observed in the CSF collected within 24 hours from the injury also reflects inflammation as the prevalent pathogenic mechanism at this stage of this progressive pathology, that will chronically evolve in neurodegeneration and demyelination. Inflammation is a major player in the so-called "secondary degeneration", that expands the initial damage, sustaining the chronic evolution of the lesion (Kwiecien et al., 2020a), and a recognized, effective target for early pharmacological intervention (Bighinati et al., 2020; Kwiecien et al., 2020b). However, being the traumatic lesion itself, such as the tissue and systemic reaction, a highly personalized process, appropriate biomarker discovery strategy should be based on serial sampling of biological fluids, from injury to stabilization. Since CSF serial sampling is ethically hardly acceptable, a further effort should be dedicated to the individual comparative analysis of blood and CSF biomarkers, also considering results in animal models (Borjini et al., 2016, 2019) "

Kwiecien JM, Dabrowski W, Dąbrowska-Bouta B, Sulkowski G, Oakden W, Kwiecien-Delaney CJ, Yaron JR, Zhang L, Schutz L, Marzec-Kotarska B, Stanisz GJ, Karis JP, Struzynska L, Lucas AR. Prolonged inflammation leads to ongoing damage after spinal cord injury. PLoS One. 2020 Mar 19;15(3):e0226584. doi: 10.1371/journal.pone.0226584. PMID: 32191733; PMCID: PMC7081990.

Kwiecien JM, Dabrowski W, Kwiecien-Delaney BJ, Kwiecien-Delaney CJ, Siwicka-Gieroba D, Yaron JR, Zhang L, Delaney KH, Lucas AR. Neuroprotective Effect of Subdural Infusion of Serp-1 in Spinal Cord Trauma. Biomedicines. 2020 Sep 23;8(10):372. doi: 10.3390/biomedicines8100372. PMID: 32977430; PMCID: PMC7598159.

Bighinati A, Focarete ML, Gualandi C, Pannella M, Giuliani A, Beggiato S, Ferraro L, Lorenzini L, Giardino L, Calzà L. Improved Functional Recovery in Rat Spinal Cord Injury Induced by a Drug Combination Administered with an Implantable Polymeric Delivery System. J Neurotrauma. 2020 Aug 1;37(15):1708-1719. doi: 10.1089/neu.2019.6949. Epub 2020 May 14. PMID: 32212901.

Borjini N, Fernández M, Giardino L, Calzà L. Cytokine and chemokine alterations in tissue, CSF, and plasma in early presymptomatic phase of experimental allergic encephalomyelitis (EAE), in a rat model of multiple sclerosis. J Neuroinflammation. 2016 Nov 15;13(1):291. doi: 10.1186/s12974-016-0757-6. PMID: 27846891; PMCID: PMC5111339

Borjini N, Sivilia S, Giuliani A, Fernandez M, Giardino L, Facchinetti F, Calzà L. Potential biomarkers for neuroinflammation and neurodegeneration at short and long term after neonatal hypoxic-ischemic insult in rat. J Neuroinflammation. 2019 Oct 28;16(1):194. doi: 10.1186/s12974-019-1595-0. PMID: 31660990; PMCID: PMC6819609.)

Round 2

Reviewer 1 Report

I would like to thank you the authors for their response, as well as for the provided supplementary materials.

I completely understand that the small sample size does not allow to conduct formal statistical testing of CSF biomarker differences across ASIA grades due to the large inter-individual variability. I believe that their responses to my comments on this could be added to the text.

“The neurological level at discharge was positively associated with MIP-1b (r=0.763), MIP-1a (r=0.644), MCP-1 (r=0.559), IL-9 (r=0.661) and IL-18 (r=0.509). The change in neurological level since admission was positively correlated to MIP-1a (r=0.508) and IL-9 (r=0.561).”

On the other hand “Some of these studies also seek to correlate biomarker level with clinical state. For example, the combination of inflammatory (IL-6, IL-8, and MCP-1) and structural neural (tau) and glial (S100β, and GFAP) biomarkers in acute SCI patients (24-h post injury) has been proposed to reflect injury severity according to the AIS scale [7, 14]. Markers which may help to distinguish between moderate and severe SCI events have also been proposed [14].”

To summarize, Spearman’s correlation coefficient showed that some early CSF biomarkers were positively associated with the neurological level at discharge and the change in neurological level since admission.

As the authors stated in their response: “As a final note, it is worth remarking that the 9 study patients are 3 ASIA grade A, 3 grade B and 3 grade D, which suggests that the correlations reported in the paper are not biased.”

Do the authors believe that the evaluated early CSF biomarkers could reflect the injury severity or not?

The authors should be congratulated for their promising findings. However, as written in the conclusion, the study of larger samples is absolutely needed to identify the early CSF biomarkers that could allow to predict long-term outcomes in patients with the same neurological level at admission.

Author Response

I would like to thank you the authors for their response, as well as for the provided supplementary materials.

We would like to thank once more the reviewer for this very positive interaction

I completely understand that the small sample size does not allow to conduct formal statistical testing of CSF biomarker differences across ASIA grades due to the large inter-individual variability. I believe that their responses to my comments on this could be added to the text.

Done, lines 249-262 of the revision 2

“The neurological level at discharge was positively associated with MIP-1b (r=0.763), MIP-1a (r=0.644), MCP-1 (r=0.559), IL-9 (r=0.661) and IL-18 (r=0.509). The change in neurological level since admission was positively correlated to MIP-1a (r=0.508) and IL-9 (r=0.561).”

On the other hand “Some of these studies also seek to correlate biomarker level with clinical state. For example, the combination of inflammatory (IL-6, IL-8, and MCP-1) and structural neural (tau) and glial (S100β, and GFAP) biomarkers in acute SCI patients (24-h post injury) has been proposed to reflect injury severity according to the AIS scale [7, 14]. Markers which may help to distinguish between moderate and severe SCI events have also been proposed [14].”

To summarize, Spearman’s correlation coefficient showed that some early CSF biomarkers were positively associated with the neurological level at discharge and the change in neurological level since admission.

As the authors stated in their response: “As a final note, it is worth remarking that the 9 study patients are 3 ASIA grade A, 3 grade B and 3 grade D, which suggests that the correlations reported in the paper are not biased.”

Do the authors believe that the evaluated early CSF biomarkers could reflect the injury severity or not?

We have included a short statement on this, lines 379-382 of the revision 2

The authors should be congratulated for their promising findings. However, as written in the conclusion, the study of larger samples is absolutely needed to identify the early CSF biomarkers that could allow to predict long-term outcomes in patients with the same neurological level at admission.